# Ionic Liquid in Phosphoric Acid-Doped Polybenzimidazole (PA-PBI) as Electrolyte Membranes for PEM Fuel Cells: A Review

**DOI:** 10.3390/membranes11100728

**Published:** 2021-09-24

**Authors:** Leong Kok Seng, Mohd Shahbudin Masdar, Loh Kee Shyuan

**Affiliations:** 1Department of Chemical and Process Engineering, Universiti Kebangsaan Malaysia, Bangi 43600, Selangor, Malaysia; leongkokseng90@gmail.com; 2Department of Petrochemical Engineering, Politeknik Tun Syed Nasir Syed Ismail, Pagoh 84600, Johor, Malaysia; 3Fuel Cell Institute, Universiti Kebangsaan Malaysia, Bangi 43600, Selangor, Malaysia; ksloh@ukm.edu.my

**Keywords:** electrolyte membrane, fuel cell, polybenzimidazole, ionic liquid, stability

## Abstract

Increasing world energy demand and the rapid depletion of fossil fuels has initiated explorations for sustainable and green energy sources. High-temperature polymer electrolyte membrane fuel cells (HT-PEMFCs) are viewed as promising materials in fuel cell technology due to several advantages, namely improved kinetic of both electrodes, higher tolerance for carbon monoxide (CO) and low crossover and wastage. Recent technology developments showed phosphoric acid-doped polybenzimidazole (PA-PBI) membranes most suitable for the production of polymer electrolyte membrane fuel cells (PEMFCs). However, drawbacks caused by leaching and condensation on the phosphate groups hindered the application of the PA-PBI membranes. By phosphate anion adsorption on Pt catalyst layers, a higher volume of liquid phosphoric acid on the electrolyte–electrode interface and within the electrodes inhibits or even stops gas movement and impedes electron reactions as the phosphoric acid level grows. Therefore, doping techniques have been extensively explored, and recently ionic liquids (ILs) were introduced as new doping materials to prepare the PA-PBI membranes. Hence, this paper provides a review on the use of ionic liquid material in PA-PBI membranes for HT-PEMFC applications. The effect of the ionic liquid preparation technique on PA-PBI membranes will be highlighted and discussed on the basis of its characterization and performance in HT-PEMFC applications.

## 1. Introduction

Rising energy demand driven by the growth of community and industries, catalyzed by the rapid depletion of petroleum-based energy sources, has initiated the interest of academicians or researchers to find new energy sources. Furthermore, energy generated from petroleum-based sources produces hazardous gases such as CO, NOx and SOx that are harmful to the environment, initiating acidic rain, depletion of stratospheric ozone and global climate change. Thus, a new type of energy producer is necessary to meet the demands while being environmentally friendly.

To date, electrical energy is the most used energy, and demands for this energy are rising with population growth. Approximately 86.4% of electrical energy is from fossil fuels, such as oil (36%), natural gas (23%) and coal (27.4%), with the remaining 14.6% generated from renewable sources [1,2,3,4]. Thus, the concept of fuel cells has emerged as an innovative energy conversion device that is considered a sustainable and environmentally friendly energy conversion procedure. Fuel cells operate via an electrochemical reaction directly converting chemical energy into electrical energy (Figure 1). This technology is considered the most promising power generation and offers several advantages including high efficiency, reliability, zero-emission, silent operation and low maintenance [5,6,7]. There are five types of fuel cells classified based on the electrolytes, namely polymer electrolyte membrane fuel cell (PEMFC), alkaline fuel cell (AFC), phosphoric acid fuel cell (PAFC), molten carbonate fuel cell (MCFC) and solid oxide fuel cell (SOFC).

PEMFC has emerged as the most promising fuel cell technology due to its high energy efficiency and power density, low emissions, compact size, light weight, fast start-up, short refueling time and low operating temperature [9,10]. There are two types of fuel cells, low- or high-temperature PEMFC, classified based on their operating temperatures. Low-temperature (LT) PEMFC usually operates below 100 °C and demonstrates exceptional performance with a maximum power density of about 500 to 100 mW cm^−2^ under H_2_/air [11,12]. However, the larger radiator volume and complex water management hinder the application of these fuel cells. In contrast, the high-temperature (HT) PEMFC operates from 100 to 200 °C, and the relatively simple design requires a simple radiator so external humidification is not necessary [13,14]. HT-PEMFC also has several advantages, such as increased electrode kinetics, higher tolerance to CO, low crossover and few by-products [15,16].

For the development of PEMFC, the most significant factor is a highly proficient polymer electrolyte membrane. Initially, perfluorosulfonic acid polymer membranes such as Nafion were used, as these materials demonstrated good conductivity, chemical and mechanical stability, as well as a higher power density [17,18]. However, this polymer fails to perform at high operating temperatures due to decreased proton conductivity and destruction of the polymer structure [19,20]. Thus, a new polymer, polybenzimidazole (PBI), is a promising alternative due to its chemical and thermal stability without humidification and low cost [21]. These polymers also contain amide or imide groups that can function as proton acceptors and react in an acidic medium [22].

To increase the conductivity of the PBI membrane, several mineral acids can be used as dopants, such as HNO_3_, H_2_SO_4_, HClO_4_ and HCl. Xing and Savadogo [23] proposed the conductivity order of mineral acids as follows H_2_SO_4_ > H_3_PO_4_ > HCIO_4_ > HNO_3_ > HCl. Even though sulfuric acid ranked higher in the list, it is not practical to apply for doped acid as more than 50% relative humidity (RH) is needed to obtain the maximum output of >0.2 S/cm at 150 °C with a doping level of 9.65 [24]. Moreover, the stability of the PBI membrane decreased rapidly in hot concentrated sulfuric acid. Thus, phosphoric acid is frequently selected as a dopant due to its higher conductivity, outstanding thermal stability and low vapor pressure at high temperatures [25]. Jones and Rozière [26] also explained that the presence of free acids in the polymer structure and H_2_PO_4_^−^/HPO_4_^2−^ anionic chains initiated higher proton conductivity of the PA-PBI polymer.

The conductivity of the PA-PBI polymer depends on the amount of phosphoric acid-doped in the polymer. However, a higher amount of phosphoric acid leads to the degradation of mechanical properties and acid leaching, thus limiting the conductivity of the pure PA-PBI. The most efficient method to overcome this problem is introducing ionic liquids into the polymer phase [9].

## 2. PBI Membrane in PEMFC

In the last decade, considerable efforts have been made to develop high-temperature (>100 °C) PEMFCs using polymer acid complexes (PACs), as they offer significant advantages in this temperature range, such as (1) improved CO tolerance, (2) enhanced efficiency, (3) avoidance of flooding by-water, (4) opportunity to use non-noble metal catalysts and (5) system simplification. Early investigation of PEMFC applied structure Nafion membrane (Figure 2) as a proton exchange membrane, and to date this material has been recognized as a reference for PEMFC [27]. This perfluorinated type of membrane was commercialized by the DuPont company and has several significant characteristics, such as high proton conductivity as well as good chemical and mechanical properties, for fuel cell operations with more than 60,000 h of operation.

However, Nafion has several drawbacks that affect the performance of fuel cells. This membrane must be firstly hydrated because the conductivity and productivity decrease quickly with decreasing RH, high methanol crossover and costly materials. Moreover, the complex design including water and heat management must be built in, as the operation of this fuel cell requires sufficient water content in the membrane to preserve the membrane conductivity and maintain the operating temperature below 80 °C.

The requirements for high-performance PEMFCs are as follows:(1)Proven stability in terms of electrochemical, chemical and thermal for fuel cell operations;(2)Higher mechanical tensile strength and sturdiness under heavy loads;(3)Higher gas separation capacity;(4)Good electrical insulation;(5)Cost-effective.

The material membranes selected have a direct impact on the storage device’s performance in a wide range of applications. Fuel cells are a promising technique to produce an environmentally friendly conversion energy system stored in a fuel. A hydrogen economy based on renewables, which includes hydrogen production, storage and power conversion, has been widely seen as a potential answer for the future of energy. Hydrogel electrolytes that are alkali-tolerant have been widely used in next-generation alkaline energy devices [28]. Preoxidized kraft lignin and poly(ethylene glycol)diglycidyether (PEGDGE) crosslinking reactions have been constructed, studied and used as quasi solid-state (QS) electrolytes in aqueous dye-sensitized solar cell (DSSC) devices, which showed a straightforward strategy for the field of sustainable photovoltaic devices [29]. The core double shell photocatalyst was a promising, magnetically separable and stable photocatalyst for long-term practical applications of photo oxidation [30]. For energy storage devices, a nanocomposite of CoSn alloy with a multishell layer structure enclosed in 3D porous carbon showed excellent performance when used as an anode for lithium-ion batteries (LIBS) [31]. A composite gel polymer electrolyte consisting of a highly cross-linked polymer matrix, containing a dextrin-based nanosponge and activated with a liquid electrolyte, exhibited good ionic conductivity at room temperature [32]. Lithium bis(trifluoromethylsulfonyl)imide (LiTFSI) on a solid polymer electrolyte (SPE) system with 30 wt.% LiTFSI doping level achieved an ionic conductivity of 3.69 × 10^−8^ Scm^−1^ at ambient temperature and 1.23 × 10^−4^ Scm^−1^ at 373 K [33,34].

Since the first successful application of poly-benzimidazole (PBI) membranes as electrolytes, PBI has been extensively explored as a proton conducting electrolyte in fuel cell applications due to its high thermal, chemical and mechanical stability and high proton conductivity [35]. PBI membranes are also highly resistant to acidic or basic conditions and have high glass transition temperatures (425–436 °C), low flammability, high energy radiation resistance and are relatively inexpensive. The most convenient characteristic of the PBI membrane is that this polymer is suitable for the high temperatures of fuel cells, as acids or alkaline groups act as a proton carrier without hydration. 

PBI monomers are linear aromatic heterocyclic macromolecules and were first introduced by Vogel and Marvel in 1961 [36] for defense and aerospace applications. Wainright et al. [37] introduced phosphoric acid-doped PBI membranes as a polymer electrolyte for HT-PEMFCs in 1995, thus initiating research in this area. Several methods were introduced to prepare this membrane, including polymerization in polyphosphoric acid (PPA), casting from methane sulfonic acid and microwave-assisted organic synthesis. To date, poly[2,2′-(*m*-phenylene)-5,5′-bibenzimidazole], also known as *m*-PBI, and poly(2,5-benzimidazole) or AB-PBI are the most common PBI membranes used for the study of HT-PEMFC [8]. *M*-PBI can be prepared via polycondensation of diaminobenzidine monomer with isophthalic acid, while AB-PBI is prepared via polycondensation of 3,4-aminobenzoic acid (DABA) either in PPA or Eaton’s reagent [3,37,38]. Figure 3a shows the chemical structure of the *m*-PBI and AB-PBI monomers, while Figure 3b,c depicts their synthesis. Figure 3d presents the synthesis of the linear and cross-linking sulfuric acid-OPBI polymer [39], and Figure 4 indicates the possible proton transfer pathway for the sulfuric acid–PBI polymer [40]. Table 1 lists examples of the preparation methods for PBI polymers and their applications.

Mekhilef et al. [50] and Zeis [51] explained that the development of HT-PEMFC was significantly influenced by PAFCs using phosphoric acid as an electrolyte. Theoretically, the proton conductivity of PBI is minimal; thus, the incorporation of secondary proton conducting materials is crucial to support ion conductivity [52]. Therefore, doping with phosphoric acid is commonly used to increase the conductivity of PBI, known as phosphoric acid-PBI or PA-PBI. This breakthrough technology was first reported by Samms et al. [53] in their study on the proton conductivity of PA-PBI via solid-state NMR, which suggested that the mobility of phosphoric acid in the PBI polymer is lower than that of free phosphoric acid. This phenomenon increased the proton conductivity of PBI polymer and maintained high thermal stability without external gas humidification. Melchior et al. [54] also added a selection of PA based on their high proton conductivity and low vapor pressure. Zeis [42] explained the mechanism of proton transfers in phosphoric acid-doped PBI membrane as shown in Figure 5.

Nevertheless, a significant drawback of PA-PBI is that high conductivity depends on the percentage of phosphoric acid loading, with a high concentration of phosphoric acid affecting the mechanical strength of PBI. In high RH conditions, leaching of PA occurs during fuel cell shutdown caused by water condensation, thus reducing the conductivity [55]. Hu et al. [56] reported that the phosphate anions could be adsorbed onto the surface of platinum, which acts as a catalyst in the fuel cell, blocking the active sites, thus causing deactivation of the platinum. Moreover, Asensio et al. [14] and Yu et al. [57] reported that the pyrolysis of PA occurred at 190 °C, thus initiating loss of proton conductivity. A high phosphoric acid content initiated several problems such as reduced mechanical strength, elution of electrolytes, corrosion of the catalyst, leaching and condensation of phosphate groups at high temperature, and formation of oligomers such as pyrophosphoric acid [3,58,59,60,61]. Extensive research has been conducted to overcome this problem with several solutions proposed, such as the introduction of silica and clay [62,63], metal carborane and metal oxides [64], phosphate salts [8], heteropolyacids [65], metal-organic frameworks [66], graphene oxide [67,68] and ionic liquids [69,70,71] (Table 2).

## 3. Ionic Liquids

In general, the term ionic liquids refers to any ionic form of liquid that has a boiling temperature below the average boiling water temperature and is liquid at ambient temperature [80,81,82,83,84], for example fused salt, molten salt and liquid organic salt. In addition, ionic liquids are non-volatile, non-flammable and exhibit good chemical and thermal stability as well as high ionic conductivity [85,86]. The specific characteristic of ionic liquids is the high ionic conductivity due to high ion density and viscosity, since they contain ions only and are 10–1000 times more viscous than water [87]. Ionic liquids have attracted considerable attention due to their new and tunable physicochemical properties, especially in electrochemical applications [40,81,82].

Ionic liquids can generally be classified into three classes, i.e., aprotic, protic and zwitterionic (Figure 6a), with each class synthesized for a specific application. Aprotic ionic liquids are a mixture of large organic cations such as pyridinium, imidazolium or phosphonium with smaller anions such as bromine, chloride, sulfate and hexafluorophosphate for inorganic anions, or bis(trifluoromethyl sulfonyl)imide for organic anions [88]. Generally, they are synthesized by alkynation of quaternization of amine groups, followed by an anion exchange reaction (Figure 5b). Aprotic liquids have no active protons in their chemical structure [89]. Protic ionic liquids are synthesized via proton transfer through a Brönsted acid and base, which act as a proton-donor and acceptor and contain exchangeable protons in their chemical structure for hydrogen bond formation (Figure 5b) [90,91]. Zwitterionic ionic liquids are prepared by adding ionic liquid compounds to surfactant systems to modify the surfactant properties (Figure 6c) [85]. Figure 7 shows the most common cations and anions for ionic liquids compounds widely used in the literature, and Figure 8 depicts several types of protic ionic liquid.

Ionic liquids can be prepared for their specific application, as both anions and cations can be incorporated; therefore, they can be used for catalysis, biocatalysis, synthetic chemistry and electrochemistry. Vekariya et al. [95] listed three generations of ionic liquids based on their applications. The first generation of ionic liquids involved the preparation of 1-alkyl-3-methylimidazolium salts by Wilkes et al. in 1982, known as tetrachloroaluminates [96]. Then, the second generation of ionic liquids was successfully developed by replacing the tetrafluoroborate ion and other anions to produce air- and water-stable ionic liquids [97], widely used as solvents for organic reactions. The third generation of ionic liquids was introduced by Davis in 2004 [98], known as task-force ionic liquid, incorporating a large group of cations, such as phosphonium, imidazolium, ammonium, pyridinium and highly diffuse anions, such as BF_4_^−^, PF_6_^−^ and CF_3_SO_3_^−^. These ionic liquids were synthesized specifically for their applications, for example, acidic chloroaluminate salts contained imidazolium and pyridinium cations for battery applications. Some reports on the usage of ionic liquids are provided in Table 3.

Due to the high conductivity and insignificant vapor pressure of the ionic liquids, these materials are stable to be employed as an additive in mid- and high-temperature PEMFC [9]. Protic ionic liquids containing N and H atoms can form a hydrogen bond network initiating the Grotthuss mechanism of conductivity, which is superior to the vehicle mechanism. Therefore, the addition of a protic ionic liquid enhanced the conductivity of the polymer electrolyte and reduced the inorganic acid dependency, such as phosphoric acid. 

## 4. Ionic Liquids in PBI Membranes

This review is focused on the effects of the ionic liquid doping techniques on the performance of PA-PBI membranes as HT-PEMFCs. The latest techniques and materials applied in ionic liquid doping PA-PBI are also discussed.

As reported by previous studies, the conductivity of the PBI membranes depended on the concentration of phosphoric acid. However, high concentrations of PA are highly corrosive, and this initiated major problems that included damaging the mechanical structure of the cells. Additionally, at higher temperatures, hydration of phosphoric acid and formation of pyrophosphoric acid oligomers reduced the conductivity of PA/PBI [102,103,104]. Therefore, several approaches were proposed, and recently, ILs were presented as a promising solution. ILs contain proton donors and acceptors in their chemical structures, which were expected to enhance the conductivity of the PBI monomers even in low PA concentrations [89].

### 4.1. Synthesis

Skorikova et al. [108] successfully developed bis(triflioromethanesulfonyl)imide-PBI membranes through direct blending to form quasi-solidified ionic liquid membranes (QSILMs). This approach was recommended for the immobilization of protic ionic liquid compounds in the polymer matrices. Immobilization of protic ILs using this technique offered several advantages such as simple procedure, low consumption of toxic organic solvents and high volume of ILs immobilization. De Trindade et al. [109] also explored the ability of ILs to improve the conductivity of the PBI membranes. They synthesized and characterized the PBI with 3-triethylammonium hydrogen sulfate (TEA) and 1-butylimidazole hydrogen sulfate as the ionic liquid compounds.

Javanbakht et al. [110] employed 1,3-di(3-methylimidazolium) to propane dibromide dicationic ionic liquid (*pr*(*mim*)_2_*Br*_2_) as the doping agent for the PA-PBI membranes. The dicationic ILs were classified as ILs as they contained two mono anions and two aromatic rings linked by alkyl chains as cations. The compound had several advantages such as higher thermal stability, glass transition temperature, melting point and proton conductivity compared to mono cationic ILs that improved the quality of the membranes for applications in HT-PEMFCs [111,112]. The investigation used melamine-based dendrimer functionalized-Santa Barbara amorphous-15 mesoporous silica (MDA-SBA-15) as the hydrophilic inorganic particles and exhibited a momentous role in the protection of the PA and pr(mim)_2_Br_2_ against water vapor, which was the by-product at the cathode after a long operation time of the HT-PEMFCs.

In another study, Compañ et al. [69] prepared a series of PA/PBI membranes that engaged different exchangeable anions in the ionic liquid to evaluate the effects of the anions and temperature on the proton conductivity of the phosphoric acid-doped PBI membranes. The study applied 1-butyl-3-methylimidazolium (BMIM) as the ionic liquid compound with several anions changed, namely chloride (Cl^−^), bromide (Br^−^), iodide (I^−^), thiocyanate (NCS^−^), bis(trifluoromethylsulfonyl)imide (NTf_2_^−^), hexafluorophosphate (PF_6_^−^) and tetrafluoroborate (BF_4_^−^) ions. The composite membranes were prepared via the casting method with 5 weight percentage (wt.%) of the ILs.

Liu et al. [113] successfully prepared a series of highly conductive cross-linked membranes with fluorine-containing polybenzimidoles (6FPBI) and 1-vinyl-3-butylimidazolium chloride base to form poly(ionic liquid) (PIL) through in situ free radical polymerization. The (PIL) technique was introduced to overcome the leaching of ionic liquid molecules problem while preserving the proton transfer pathway. The PIL was a series of repeating monomers bonded with ionic liquid molecules, either anionic or cationic species of ILs [114,115]. Liu et al. [116] also prepared a series of cross-linked fluorine-polybenzimidazole (6FPBI) membranes with the addition of a cross-linked polymeric ionic liquid for HT-PEM applications. Instead of using linear ionic liquid compounds, they suggested the application of cross-linked polymeric ionic liquid compounds, as the polymeric ILs had several advantages such as providing a continuous and fast pathway for proton transfer with the aids of anions in the polymeric ILs that generally act as proton acceptors. Additionally, multiple proton transport channels were created with the incorporation of polymeric ILs in the polymer matrix, thus preventing the leakage of the ionic liquid compounds.

In a recent study, Gao et al. [117] researched the preparation and characterization of a series of PBI with hyperbranched cross-linked membranes with imidazolium groups that acted as the ionic liquid compounds. Generally, branched polymers showed significant advantages such as good oxidative stability and adsorbed more PA compared to linear polymers. However, the loss of mechanical properties was also observed in the polymers [118]. Thus, the cross-linking method was a promising solution for the improvement of branched polymers’ mechanical strength, as the cross-linkable compounds toughen the interactions among the polymer chains. Additionally, selection of the imidazolium group was crucial, as the group had a conjugated ring structure that accommodated more space for phosphoric acid in the polymer chains, and the delocalization and formation of hydrogen bonds stabilized the adsorbed phosphoric acid, which prevented leakages [119].

### 4.2. Effect of Ionic Liquids on PA/PBI Membrane Performance 

Based on the findings by Liu et al. [112], the formation of PIL was vital to promote proton transfer and to improve the mechanical properties of the PA-PBI membranes. The incorporation of PIL initiated better proton conductivity, more than 76.9% increment at 170 °C, compared to the pristine 6FPBI membranes. Moreover, phosphoric acid’s stability was increased by about 73.1% at 160 °C operating temperature. Epoxy groups in the PIL played a significant role in PA retention as the groups acted as cross-linkers and formed cross-linked networks and prevented leaking of the PA. The increased stability of the PA was expected due to the incorporation of the PIL with dihydrogen phosphate ion (H_2_PO_4_^−^). Figure 9 illustrates the synthetic process of 6FPBI and 6FPBI-PIL membranes.

The most significant observation from a study by Liu et al. [115] was that PA retention for this membrane was improved, which prolonged conductivity and stability, even at a longer time of PA doping. The cross-linked membranes also displayed better chemical and oxidative stability and good mechanical properties compared to linear membranes. Moreover, extremely high PA doping levels were achieved, therefore increasing the ionic conductivity of the membranes without any leaking detected. At 170 °C, the proton conductivity of the 6FPBI-cPIL reached about 0.106 S/cm with a doping level of 27.8. Figure 10 depicts the schematic process for the synthesis of (a) cross-linkable polymeric ionic liquid compounds and the (b) preparation of 6FPBI-cPIL. A cross-linkable polymeric liquid compound for this study was prepared via the free radical polymerization of 1-vinyl-3-butylimidazolium trifluoromethanesulfonyl imide ([ViBuIm][TFSI]) with allyl glycidyl ether.

The conductivity of QSILMs prepared by Skorikova et al. [108] achieved about 30–60 mS cm^−1^ at 180 °C after doping with phosphoric acid compared to the zero IL-PBI membranes, which only achieved less than 10 mS cm^−1^. This research also suggested a 1:1 ratio of bis(triflioromethanesulfonyl) imide-PBI membrane is the best performing ionic liquid PBI membrane, as this polymer reached a power density of about 0.32 Wcm^−2^ at 200 °C and 900 mAcm^−2^. Therefore, ionic liquid compounds have an important role, especially by increasing the conductivity of the PA-PBI membrane by improving retention of doped PA in the membrane with no leaking. The impregnated catalyst layer of the gas diffusion electrode with protic ionic liquid exhibited better stability for long-term use (100 h of operation at 200 °C) compared to phosphoric acid alone. Moreover, this research also suggested the application of fluorescence microscopy for the structural investigation of the PBI membrane and ionic liquid distribution. Generally, PBI contained fluorescence-active molecules in a broad wavelength range [120,121]; thus, the application of fluorescence microscopy will facilitate the analysis of PBI film morphology.

The doping of ionic liquid compounds initiated higher oxidative stability and proton conductivity compared to the non-doped PBI membranes even at higher temperatures and percentages of relative humidity (%RH). TEA ILs produced higher conductivity and oxidative stability compared to BIm ILs, which was caused by the presence of SO_3_H groups at the cations and SO_4_H groups at the anions. The H^+^ generated at the anions and cations increased the number of protons at the membranes, which resulted in improved conductivity compared to the BIm compounds [109]. Javanbakht et al. [110] explored the application of dicationic ILs in the preparation of PA/PBI membranes. From the results of the study, the prepared poly(2,5-benzimidazole)-dicationic ionic liquid (ABPBI-DIL_4_) showed higher proton conductivity and easier proton exchange, as a lower voltage drop was observed compared to the non-doped ABPBI membranes at a high operating temperature of 180 °C. The observation proved the ability of the ILs to increase the proton conductivity and initiate oxidative stability of the PBI membranes. The oxidative stability was increased with increased PBI content, and the higher conductivity of the membranes was parallel with the increased ILs percentage. Moreover, the formation of hydrogen bonds between acid protons of the ILs cations with the amine group of the PBI prevented the leaching of the ILs.

Compañ et al. [69] observed that the application of ILs as fillers improved the mechanical properties of the PBI membranes, which were caused by the interaction of the polymer matrix and the ionic liquid compounds. The casting technique successfully produced membranes with better thermal, mechanical and oxidative stability, which made the membranes suitable for fuel cell applications. The PBI that contained BMIM-BF4 achieved 94 mS/cm conductivity at 200 °C compared to non-doped PBI membranes, which were observed at about 0.71 mS/cm. Higher conductivity might be initiated by the formation of hydrogen bond networks between the ionic liquid compounds and the PA molecules which were distributed in the polymer matrix. Additionally, the study found that polarity and hygroscopicity were the two significant factors that described the difference in the conductivity of exchangeable anions. Therefore, the properties and quality of the PBI membranes could be modified by selecting suitable anionic molecules in the ionic liquid compounds.

Figure 11 shows the schematic diagram for the preparation of PBI composite membranes in the research by Compañ et al. [69]. The most significant finding from the literature was that the activation energy (E_act_) related to conductivity was dependent on the types of anions and obeyed the trend E_act_(NTF_2_^−^) < E_act_(Cl^−^) < E_act_(BF_4_^−^) < E_act_(NCS^−^). Consequently, anion selection was showed to be vital to improve the conductivity of the prepared ionic liquid PA-PBI membranes. The NCS anion showed the highest conductivity due to the presence of N and C atoms, which initiated more hydrogen bonds with the PA and PBI monomers, thus creating extra pathways for proton transfer. The activation energies for all membranes range from 65 to 84 kJ/mol, suggesting that the ionic conduction in these membranes primarily occurs through the vehicle-type mechanism.

Hyperbranched cross-linker ImOPBI-x membrane had outstanding oxidation stability, higher proton conductivity and satisfactory tensile strength, which meets the requirements for HT-PEMFC applications. This research also proved the ability of imidazolium groups to adsorb more phosphoric acid as well as stabilizing these molecules via delocalization and formation of hydrogen bonds due to the conjugated ring structure. Fuel cells equipped with this membrane showed a power density of 638 mW/cm^2^ at 160 °C and had good durability under a hydrogen/oxygen atmosphere, proving their ability in anhydrous proton exchange membrane applications [117]. Figure 12 shows the schematic steps for the preparation of hyperbranched cross-linker ImOPBI-x membrane, and after adsorption of phosphoric acid, imidazolium group ionic liquids hold more phosphoric acid in the polymer structure, thus increasing the membrane conductivity.

Mishra et al. [122] explained the mechanism of proton conductivity of doped phosphoric acid in the presence of ionic liquid compounds. They synthesized an AB-PBI membrane with 1-(3-trimethoxysilylpropyl)-2-methylimidazolium tetrafluoroborate to act as ionic liquid compounds. The formation of hydrogen bonds between hydrogen molecules from phosphoric acid with nitrogen molecules at PBI and ionic liquid structures helps the movement of proton along the polymer chain, thus initiating the conductivity of the polymer. Even though less phosphoric acid is adsorbed in the polymer chain, protonation still occurred as the presence of ionic liquid compounds held the phosphoric acid molecules via the formation of hydrogen bonds. Liu et al. [123] also suggested the possible proton transfer in the ionic liquid-based PBI membrane. The H-N bond from the ammonium cation of [dema][TfO] might interact with the C=N bond of PBI and proton transfer from H-N bond to C=N to C=N amine. A high content of [dema][TfO] provides more free protic ionic liquid, which acts as a proton conductor to improve proton transfer. Figure 13 depicts the mechanism of proton conduction in the phosphoric acid-doped AB-PBI membrane with the presence of ionic liquid compounds.

Among the protic ionic liquids, the most commonly explored is N,N-diethyl-*N*-methylammonium triflate ([dema][TfO]) (Figure 14) [124]. Sen et al. [125] and Niu et al. [71] concluded that PBI-[dema][TfO] membrane had higher ionic conductivity and stability compared to the other ionic liquids, performing better and generating 144 and 62 mW/cm^2^ at 125 °C and 250 °C [71]. Pant et al. [124] conducted molecular-dynamic (MD) simulations of [dema][TfO] doped PBI, finding higher membrane conductivity with increased doped ionic liquids, both for simulations and experimental findings. This may be due to the formation of well-developed ionic channels and the presence of free mobile ions. Table 4 lists the conductivity of different protic ionic liquids at their operating temperature.

Based on the review of several studies, it can be concluded that the presence of the ionic liquids in the phosphoric acid-doped PBI membrane has significant effects, especially on the membrane conductivity, mechanical strength and stability. Table 5 lists the ionic liquid compounds that have been applied in PA-PBI membrane in the literature and outcomes.

## 5. Future Prospects

To date, the development of fuel cell technology is wide open, and this niche area currently attracts various researchers, including industrial players, to study effective fuel cells. The sustainable fuel cell must include the reliable cost of manufacture, which is related to the production, storage, transportation and distribution of hydrogen, and maximum output generated [151]. The effective design of PBI-based membranes doped either with organic or inorganic materials must be finalized; thus, higher performance and durability of the fuel cell will be achieved.

Most importantly, if phosphoric acid is required in the PBI-based membrane, the leaking problem must be resolved. Currently, the application of polymeric ionic liquids in the fabrication of PA-PBI membrane seems a promising solution, as polymeric ionic liquid forms additional networks in the membrane, reinforcing the membrane structure, thus enhancing the mechanical property of the PEM. However, there are several considerations for upscaling, such as low-cost production, simple process, optimum stoichiometric ratios, reaction conditions, purification, recovery and production of films [152].

## 6. Conclusions

The addition of ionic liquid compounds may significantly improve the performance of PA-PBI membranes for fuel cell application. Aside from proton donors, the ionic liquid compounds can also act as retention agents to prevent the leaching of phosphoric acid in the PA-PBI membrane. Moreover, the mechanical and thermal stability and proton conductivity of the PA-PBI membrane can be modified via the selection of the anion or cation compounds in the ionic liquid structure to achieve the perfect and workable polymer electrolyte for the HT-PEMFC application. The proton conductivity of ionic liquid PA-PBI is initiated by the formation of a hydrogen bond network between ionic liquid molecules and nitrogen atoms in the PBI membrane, thus increasing the specific conductivity and preventing loss of voltage in the fuel cells. Future work in this area is necessary to explore other preparation routes or different ionic liquid compounds, especially poly(ionic liquids), as this area is still relatively new for fuel cell applications. Strengthening the backbone of polymers is needed to increase the mechanical strength of fuel cells, thus prolonging the durability and making the fuel cell industry more economical.

## Figures and Tables

**Figure 1 membranes-11-00728-f001:**
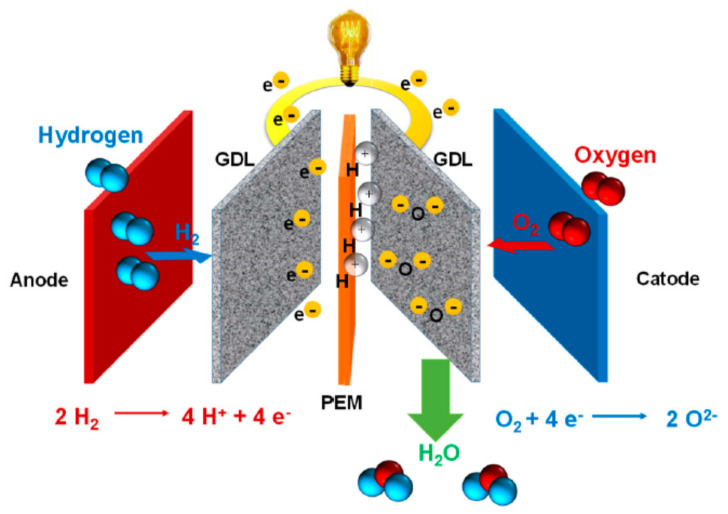
Schematic flow and chemical equation of a polymer electrolyte membrane fuel cell [8].

**Figure 2 membranes-11-00728-f002:**
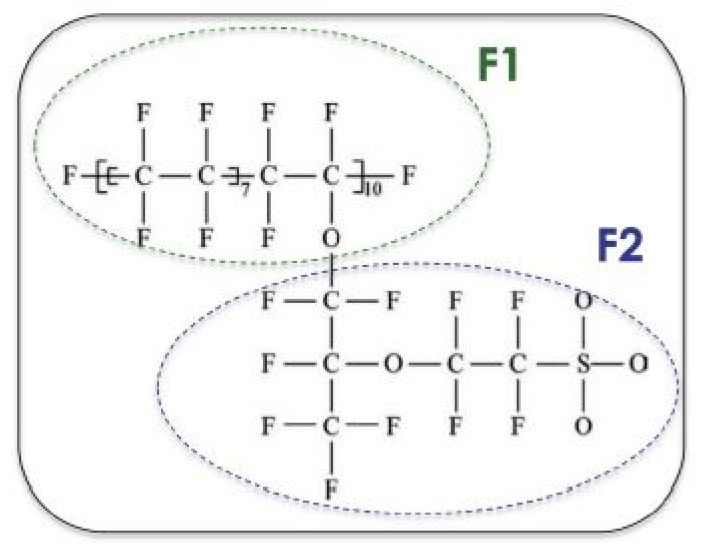
Structure of the Nafion membrane. F1 and F2 are the labels of fluorine atoms in the backbone and branching [27].

**Figure 3 membranes-11-00728-f003:**
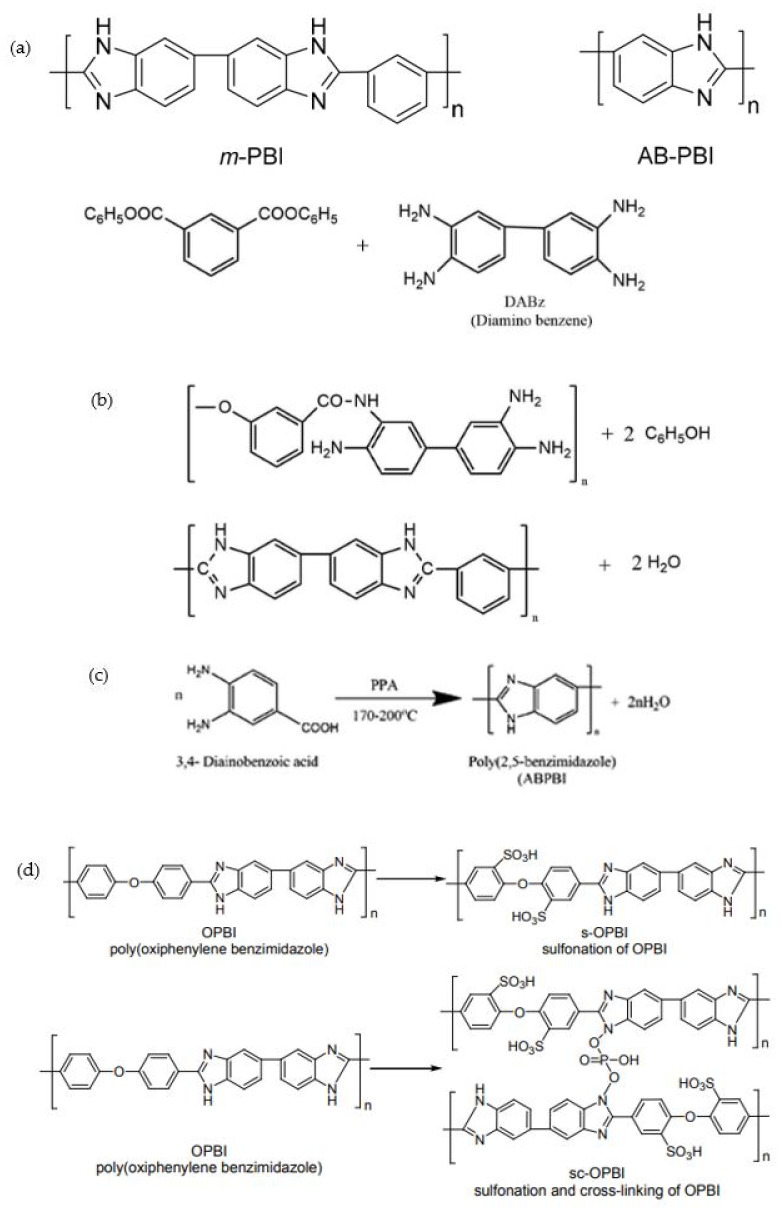
Chemical structure of (**a**) monomers for the most applied PBI polymers [8]. (**b**) Synthesis of *m*-PBI [3] and (**c**) AB-PBI polymers [41]. (**d**) Synthesis of the linear and cross-linking sulfuric acid-OPBI polymer [39].

**Figure 4 membranes-11-00728-f004:**
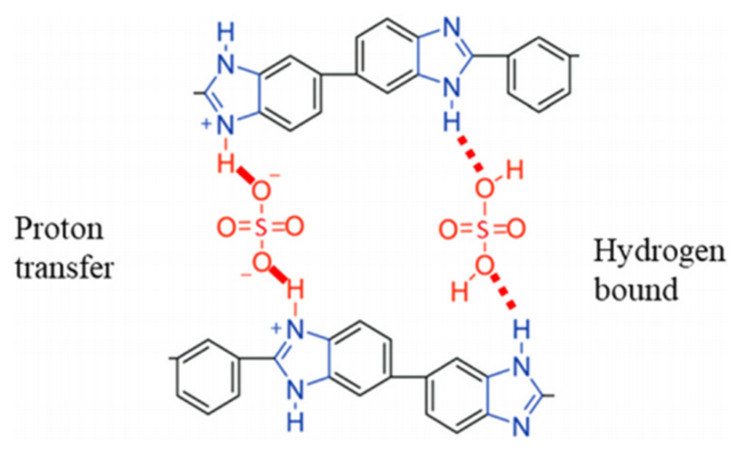
Possible proton transfer pathway for the sulfuric acid-PBI polymer [40].

**Figure 5 membranes-11-00728-f005:**
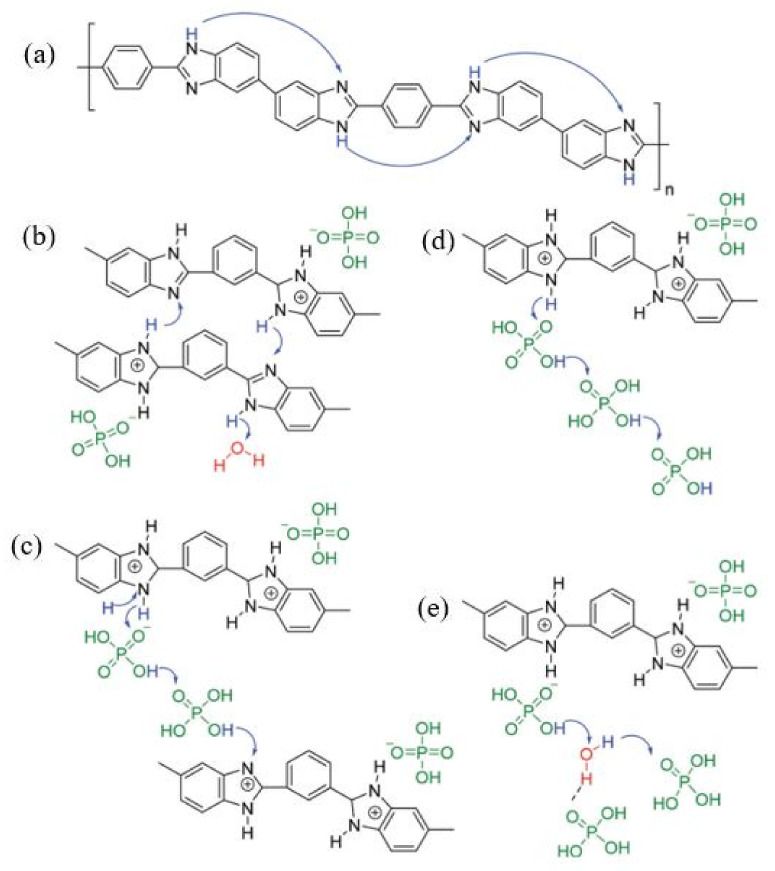
Mechanism of proton transfer in the phosphoric acid-doped PBI membrane. (**a**) Chemical structure of PBI, (**b**) PA protonated PBI with no free acid molecules, (**c**) proton transfer along with acid-benzimidazole acid, (**d**) proton transfer along acid-acid and (**e**) proton transfer along acid-H_2_O [51].

**Figure 6 membranes-11-00728-f006:**
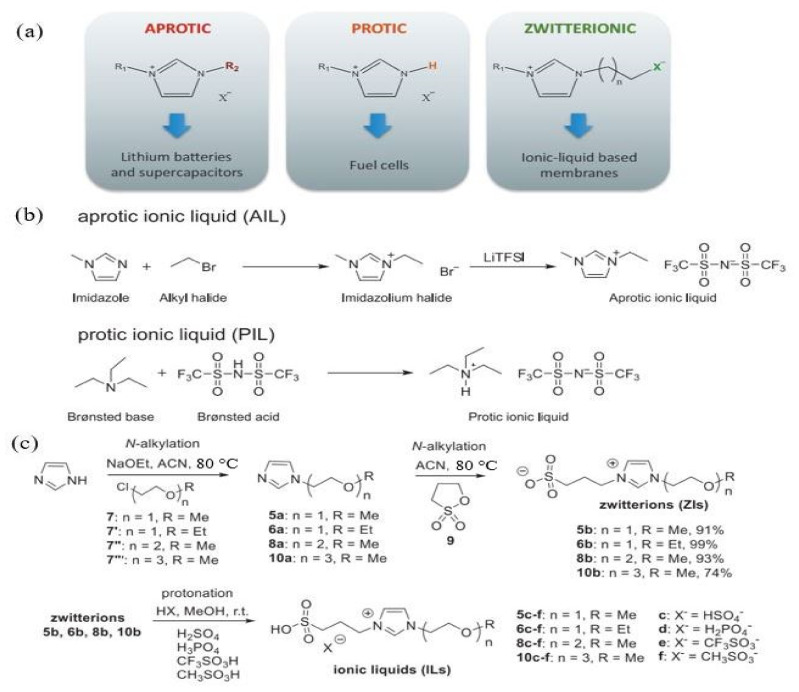
(**a**) Class of ionic liquid compounds and their specific applications [84]; (**b**) preparation of aprotic and protic ionic liquids [89]; (**c**) synthesis of zwitterion ionic liquids [92].

**Figure 7 membranes-11-00728-f007:**
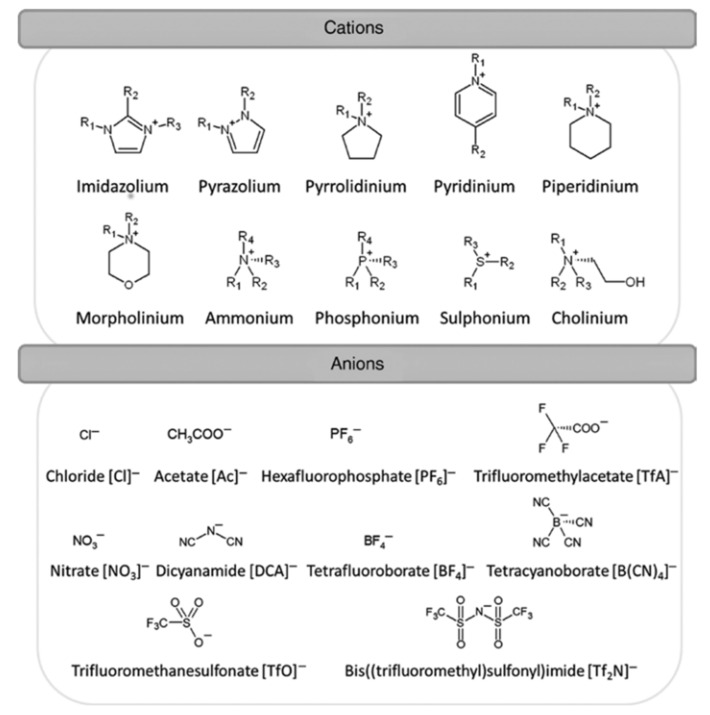
Chemical structure of the most common cations and anions widely used for ionic liquid compounds [93,94].

**Figure 8 membranes-11-00728-f008:**
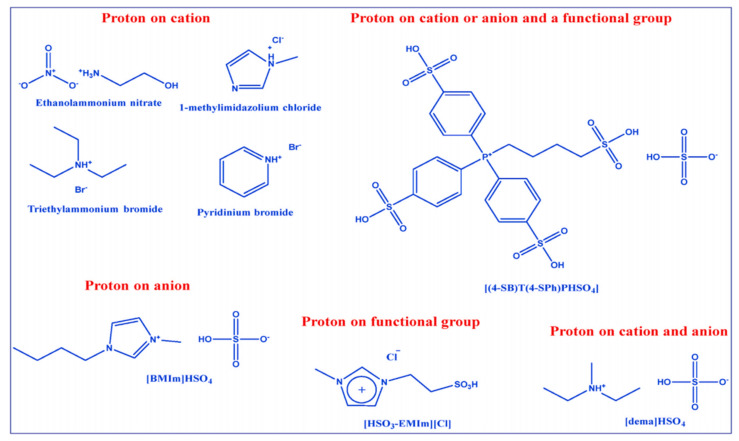
Types of protic ionic liquids [9].

**Figure 9 membranes-11-00728-f009:**
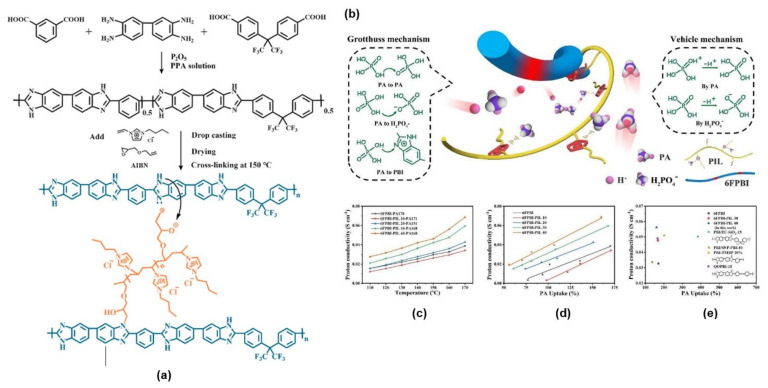
(**a**) Schematic diagram for the preparation of the 6FPBI membrane, (**b**) proton transfer mechanism, (**c**) proton conductivity of all membranes with 151–171% PA uptake, (**d**) PA uptake at 170 °C and (**e**) comparison of proton conductivity [113].

**Figure 10 membranes-11-00728-f010:**
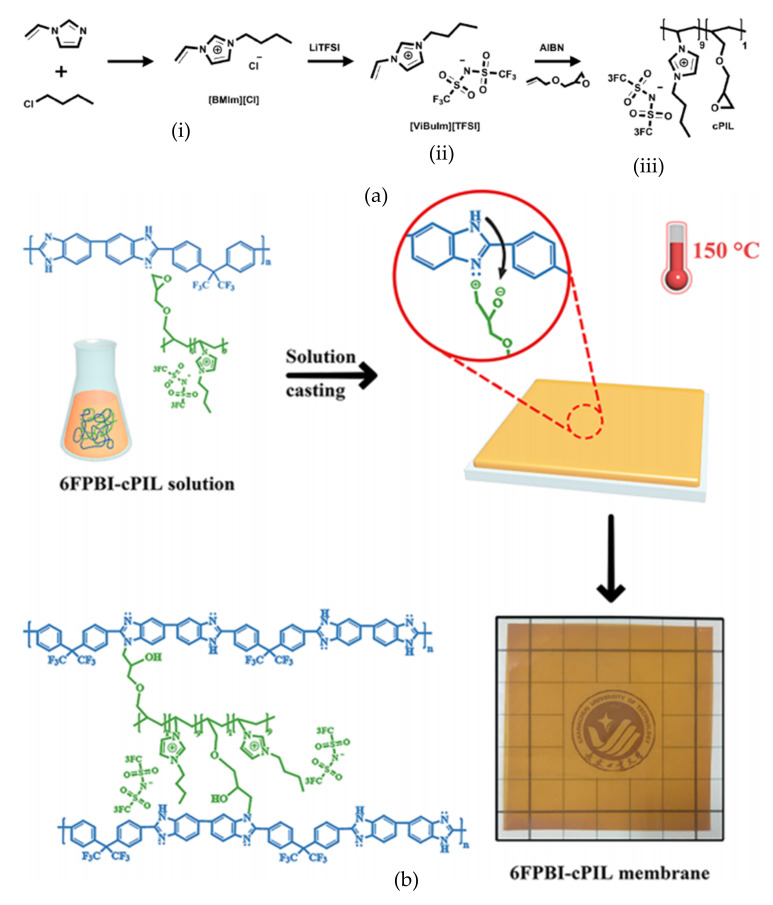
Schematic preparation of 6FPBI-cPIL membrane with (**a**) preparation of cross-linkable polymeric ionic liquid via (**i**) synthesis of [ViBuIm]Cl, (**ii**) anion exchange reaction to form [ViBuIm][TFSI] and (**iii**) free radical polymerization of [ViBuIm][TFSI] with allyl glycidyl ether to form cPIL. (**b**) Preparation of 6FPBI-cPIL membrane via a solution casting method [116].

**Figure 11 membranes-11-00728-f011:**
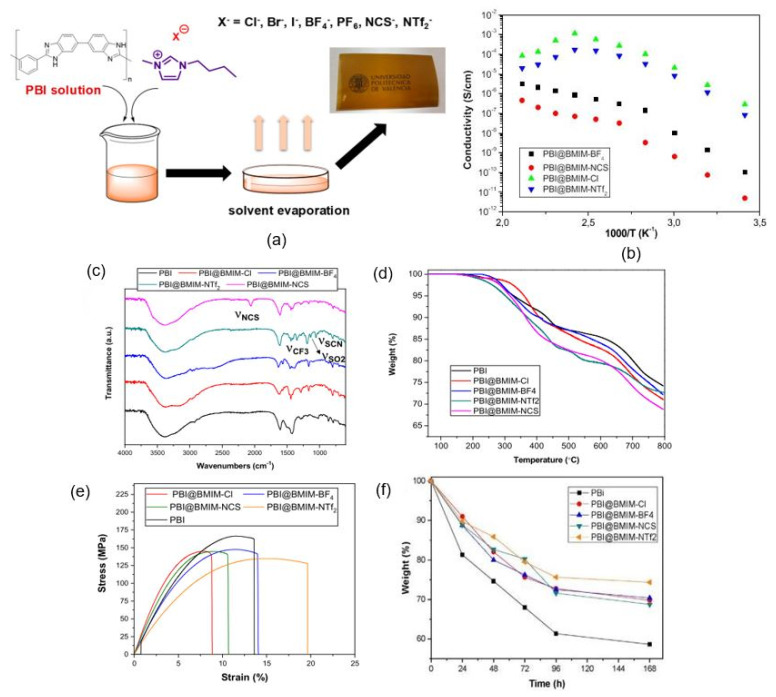
(**a**) Schematic preparation of PBI membrane containing BMIM ionic liquid compounds with different anions, (**b**) conductivity test, (**c**) FT-IR results, (**d**) TGA results, (**e**) stress–strain curves and (**f**) Fenton’s test [69].

**Figure 12 membranes-11-00728-f012:**
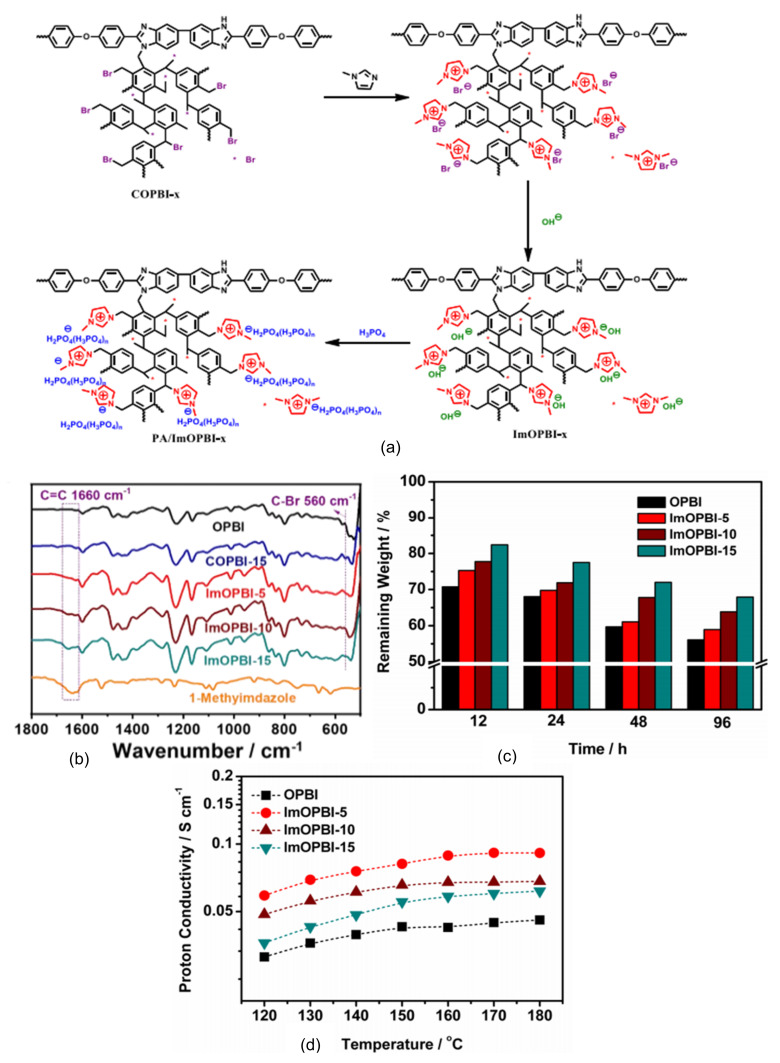
(**a**) Preparation of hyperbranched cross-linker ImOPBI-x, (**b**) FT-IR results, (**c**) phosphoric acid retention ability and (**d**) proton conductivity [117].

**Figure 13 membranes-11-00728-f013:**
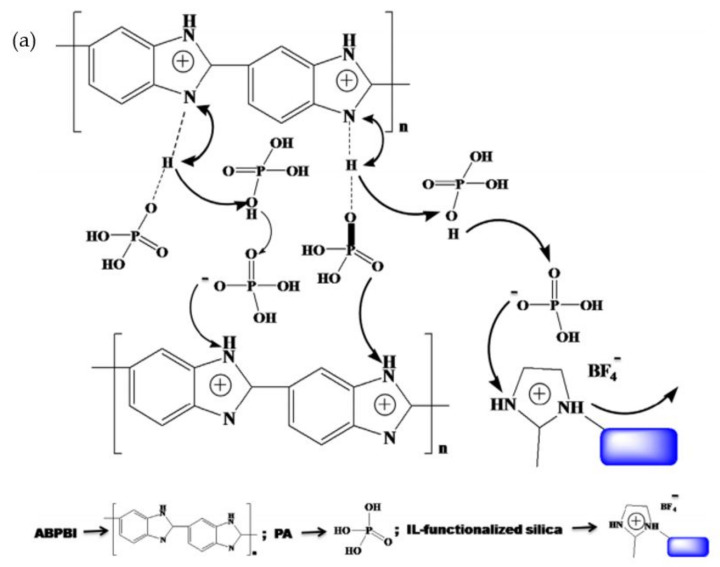
Mechanism of proton conduction in phosphoric acid-doped AB-PBI membrane with ionic liquid compound suggested by (**a**) Mishra et al. [122] and (**b**) Liu et al. [123].

**Figure 14 membranes-11-00728-f014:**
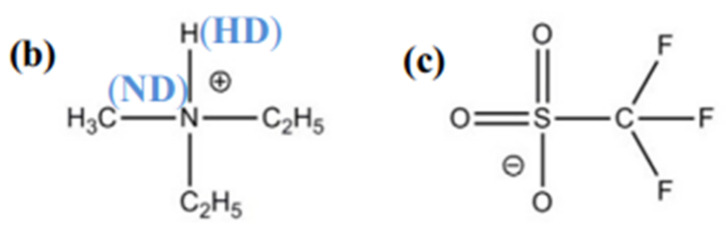
Chemical structure of N,N-diethyl-*N*-methylammonium triflate ([dema][TfO]) [124].

**Table 1 membranes-11-00728-t001:** Preparation methods for PBI polymers and their applications.

PBI Polymer	Monomer	Method	Application	Remarks	Ref.
Sulfonated polybenzimidazole	3,3′,4,4′-tetraaminobiphenyl (TAB)	Polymerization	Redox flow batteries	Sulfonated polybenzimidazole (s-PBI) gel membrane showed high conductivity (>240 mS/cm) and low degradation during in-cell testing	[42]
*m* and *p*-PBI	3,3′-diaminobenzidine	Polymerization	Heavy metal absorbent	*p*-PBI polymer indicated better performance compared to *m*-PBI	[43]
*m* and AB-PBI/ZnO	3,4,-diaminobenzoic	*m*-PBI/ZnO—doping ZnO in DMAc solution with *m*-PBI-based powder AB-PBI/ZnO—in situ polymerization of 3,4-diaminobenzoic with zinc nitrate.	Photocatalytic degradation of organic dyes	*m*-PBI/ZnO performed better as a photocatalyst compared to AB-PBI/ZnO	[44]
PBI	3,3′-diaminobenzidine	Cross-linked polymerization of 4,4-dicarboxydiphenyl ether with 3,3′-diaminobenzidine in phosphorus pentaoxide/methanesulfonic acid	Nanofiltration membranes	The cross-linked PBI membranes exhibited more than 99% retention of Rose Bengal (RB) dye in DMF solvent	[45]
*m* and *p*-PBI	3,3′,4,4′-tetraaminobiphenyl (TAB)	Polymerization of TAB with polyphosphoric acid	Electrochemical hydrogen separation	PBI membranes were used as polymer electrolytes for the EHS device Prepared PBI membranes operate the EHS device even in high CO, producing 99.6% purity of hydrogen products with very high power efficiencies (>72%)	[46]
PBI-mixed matrix membranes	3,3′-diaminobenzidine (DAB)	Polymerization of DAB with polyphosphoric acid, followed by a casting process to dope different percentages of zeolite	Gas separation	PBI-MMMs more favorable for separation of CH_4_	[47]
Asymmetric PHI-HFA hollow fiber	4,4′-(hexafluoroisopropylidene)bis(benzoic acid)	Dry-jet wet spinning	Hydrogen-selective membrane	Prepared PBI-HFA had higher permselectivities of H_2_/N_2_ and H_2_/CO. Higher hydrogen flow rates were recorded compared to other gas	[48]
*m*/*p*-PBI copolymer and *p*-PBI homopolymer	3,3′,4,4′-tetraaminobiphenyl (TAB)	PPA sol-gel process	Electrochemical hydrogen separation	The proton conductivity and mechanical properties of the polymer depend on the final membrane composition *m*/*p*-PBI copolymer exhibited higher creep resistance compared to homopolymer *p*-PBI *m*/*p*-PBI copolymer showed long-term durability and cell recoverability	[49]

**Table 2 membranes-11-00728-t002:** Several examples of fillers used in the preparation of PBI polymers.

Filler	Outcomes	Ref.
CaTiO_3_	Prepared membranes with a higher content of nano CaTiO_3_ show higher conductivity and good oxidative stability 15% nanoCTO-PBI have 32.7 mS/cm conductivity, while 5% nanoCTO-PBI indicated conductivity of 20.2 mS/cm The power density and current density of 10% nanoCTO-PBI membrane at 0.6 V and 160 °C are approximately 251.4 mW/cm^2^ and 419 mA/cm^2^	[72]
Cerium triphosphonic-isocyanurate (Ce-TOPT)	Addition of Ce-TOPT proton conductor to overcome acid leaching problems TOPT contains three –PO_3_H_2_ preventing water-insolubility of the membrane Ce-TOPT/PBI showed good mechanical properties, proton conductivity, durability and membrane selectivity Conductivity of the c-mPBI/Ce-TOPT(50) reaches 0.0885 S/cm for 50% RH, 0.125 S/cm for 100% RH and 0.0363 S/cm in anhydrous conditions Proton conductivity loss about 4.6% after 48 h water-washing	[73]
Sulfophenylated TiO_2_	Metal oxide acts as filler and cross-linker Ionic cross-linked system changes to covalently cross-linked system via thermal curing 6c-sTiO_2_-PBI-OO (6 wt.% TiO_2_) showed the highest uptake of phosphoric acid (392 wt.%) and proton conductivity of 98 mS/cm at 160 °C In fuel cell applications, a power density of 356 mW/cm^2^ obtained by the PBI membrane with filler, 76% higher compared to non-filler PBI membrane	[74]
Phosphonated graphene oxide	76.4 × 10^−3^ S/cm proton conductivity is achieved at 140 °C under anhydrous conditions Conductivity is more stable with the addition of PGO to the membrane Strong correlation between PGO content and stability of acid content In fuel cell applications, a power density more than 359 mW/cm^2^ at 120 °C under anhydrous conditions, 75% more than non-PGO PBI membrane	[75]
Graphene oxide	High power density is obtained from GO/PBI, about 546 W compared to non-GO (468 W) Hydrophilic structure of GO reduced acid stripping in the membrane, improving proton conductivity	[76]
Imidazole grapheme oxide (ImGO) and grapheme oxide	Addition of ImGO improved physicochemical properties and higher proton conductivity Addition of 0.5 wt.% ImGO enhanced tensile strength (219.2 MPa) compared to 0.5 wt.% GO (215.5 MPa) and pure PBI membrane (181.0 MPa) 77.52 mS/cm of proton conductivity is obtained by ImGO/PBI ImGO provides an additional effective proton transfer pathway	[68]
Multiwall carbon nanotubes (MWCNTs)	Pt-PBI/MWCNT shows more durability compared to Pt/C and Pt/MWCNT after the 1000th cycle of voltammetry The peak current and power density of Pt-PBI/MWCNT are lower than commercial grade Pt/C, caused by the nanotube-polymer blocking the framework catalytic areas Power density of Pt-PBI/MWCNT slightly increased at elevated temperatures, reaching 47 mW/cm^2^ at 180 °C Addition of MWCNTs improved the durability of the Pt catalyst	[77]
Zeolitic imidazolate framework	Proton conductivity increased with increasing temperature A mixture of ZIF-67 and ZIF-8 showed higher proton conductivity, about 9.2 × 10^−2^ S/cm, indicating a synergistic effect on proton conductivity	[78]
UiO-66 metal-organic framework	Introduction of UiO-66 metal-organic framework constructed channels for proton transfer Increased UiO-66 caused the tendency for decreasing phosphoric acid loading and swelling ratios UiO-66 also increased mechanical properties, long-term stability and enhanced PA retention CBOPBI-40% UiO-66 achieved proton conductivity about 0.1 S/cm and 607 mW/cm^2^ power density at 160 °C with gas humidification	[79]

**Table 3 membranes-11-00728-t003:** Some reports on the usage of ionic liquids.

Ionic Liquids	Application	Remarks	Ref.
1-methylimidazolium	Biopolymer solvent for preparation of collagen-alginate hydrogels	Ionic liquid showed a decent potential for the preparation of collagen and alginate hydrogels	[99]
1,1′-(5,14-dioxo-4,6,13,15-tetraazaoctadecane-1,18-diyl) bis(3-(sec-butyl)-1H-imidazol-3-ium) bis((trifluoromethyl)-sulfonyl) imide	Electrolyte additive in lithium-ion battery	A novel dicationic room temperature ionic liquid showed a remarkable potential to subsitute conventional organic carbonate electrolyte mixture Prepared ionic liquid was safer to use at high operation temperature with no degradation, enhanced battery life, good cycling performance and Coulombic efficiency with better discharge capacities	[100]
1-ethyl-3-methylimidazolium acetate	Solvent	Ionic liquid was employed as a solvent to dissolve chitosan before coating the surface of the chitosan hydrogel beads Simple but effective method for cellulose coating compared to other organic solvents	[101]
1-butyl-3-methylimidazolium bromide	Co-solvent for preparation of h-MoO_3_	Ionic liquid is significant for the development of hollow rod-shaped morphology h-MoO_3_	[102]
[SO_3_H-Pyrazine-SO_3_H] Cl	Catalyst	Ionic liquid was prepared accordingly to apply as a catalyst for preparation of xanthenediones and 3,4-dihydropyrimidin-2(1*H*)-ones under solvent-free conditions Several advantages were achieved, including simplicity in preparing and handling the catalyst genenrality, easy workup procedure, high yields, short reaction times, catalyst can be reused and solvent-free conditions	[103]
1-allyl-3-methylimidazolium chloride	Adsorbent for determination of oxytetracycline in milk sample	A simple, effective, sensitive and environmentally friendly method for determination of oxytetracycline in milk sample via SPME-CE	[104]
1-(4-sulfonate)-butyl-3-vinylimidazolium	Catalyst for esterification pre-treatment	A task-specific zwitterion monomer was synthesized for production of polyzwitterion support for phosphotungstic acid grafting Phosphotungstic acid was able to immobilize in the polymer support through chemical effects, and catalytic performance is superior due to reusability of the catalyst	[105]
1-hexadecyl-3-vinylimidazolium bromide	Chemical agent for oil recovery	Synthesized polyionic liquid (PIL) showed good salt tolerance behavior, thermal stability and wettability alteration ability Core flooding with PIL enhanced >30% of oil recovery after water flooding	[106]
1-butyl-3-methylimidazolium chloride	Green solvent and porogen	Ionic liquid assisted the development of π-π stacking and Van Der Waals interaction toward agglomeration of the grapheme oxide sheet Ionic liquid medium also acted as a porogen to create higher surface area of composite with better active site	[107]

**Table 4 membranes-11-00728-t004:** List of conductivity for different protic ionic liquids at their operating temperature.

Protic Ionic Liquids	Conductivity (mS/cm)	Temperature (°C)	Ref.
Pyrrolidinium nitrate	50.1	25	[126]
Pyrrolidinium hydrogen sulfate	6.8	25	[126]
Pyrrolidinium formate	32.9	25	[126]
Pyrrolidinium acetate	5.9	25	[126]
Pyrrolidinium trifluoroacetate	16.4	25	[126]
Pyrrolidinium octanoate	0.8	25	[126]
Pyrrolidinium bis(trifluoromethanesulfonyl)amide	39.6	130	[127]
7-methyl-1,5,7-triazabicyclo[4.4.0]dec-5-ene bis(trifluoromethanesulfonyl)imide	1.54	30	[128]
Diethylmethylammonium trifluoromethanesulfonate	43	120	[129]
Diethylmethylammonium hydrogen sulfate	1.10	30	[130]
Diethylmethylammonium bis(trifluoromethanesulfonyl)amide	7.40	30	[130]
Trioctylammonium triflate	0.0303	25	[131]
Benzimidazolium bis(trifluoromethanesulfonyl)imide	8.3	140	[132]
3-(1-butyl-1H-imidazol-3-ium-3-yl)propane-1- sulfonate 1,1,1-Trifluoro-N-(trifluoromethylsulfonyl) methanesulfoneamide	1	100	[133]
Morpholinium formate	9.92	60	[134]
N-methylmorpholinium formate	16.77	60	[134]
N-ethylmorpholinium formate	12.17	60	[134]
Methylimidazolium bis(trifluoromethanesulfonyl)imide	7.23	25	[135]
1-methyl-pyrazole N,N- bis(trifluoromethanesulfonyl)imide	12	90	[136]
1H-1,2,4-triazole/methanesulfonic acid	149	200	[137]
Isobutyramide trifluoromethanesulfonate	32.6	150	[138]
2,3-dimethyl-1-ethylimidazolium dihydrogenphosphate	70	120	[139]
Trifluoroacetic propylamine	30	180	[140]
Triethylammonium triflate	31	130	[141]
1-ethyl-3-methylimidazolium hydrogen sulfate	16	85	[142]
N-butylguanidinium tetrafluoroborate	180	180	[143]

**Table 5 membranes-11-00728-t005:** List of ionic liquid compounds used in Pa-PBI membranes.

Ionic Liquid Type	Outcomes	Conductivity	Ref.
2-bromo-N,N-dimetylethanamine	Ionic liquid compound was protonated using trifluoromethanesulfonic acid and phosphoric acid Application of prepared membrane as a catalyst binder increased coverage and desorption kinetics of oxygenated species on the catalyst surface, thus improving electrode reaction kinetics and catalytic activity of Pt/C catalyst.	-	[144]
Diethylmethylammonium trifluoromethanesulfonate ([dema][TfO]) 1-ethyl-3-methylimidazolium trifluoromethanesulfonate ([emim][TfO]) 1-methylimidazolium bis(trifluoronethane sulfonyl)imide ([C1Im][NTf_2_]) 1-(2-Hydroxyethyl)-3-methylimidazolium bis(trifluoromethane sulfonyl)imide (HOemim][ NTf_2_])	PBI-[dema][TfO] showed better proton conductivity compared to other PBI-ionic liquid composites Proton can transfer from the H-N bond at the ammonium cation to C=N to C=N to amine Free amine and diethylamine continuously accept proton at cathode and transport along the hydrogen bond in the PBI-[dema][TfO] membrane Low activation energy of proton conduction for [dema][TfO] is also a significant factor for higher proton conductivity	108.9 mS/cm at 250 °C	[71]
Poly(vinylimidazolium) bromide (PVImBr)	Better interfacial properties, greater tensile strength, storage modulus, acid loading, proton conductivity and low acid leaching	0.25 S/cm at 160 °C	[145]
2-sulfoethylmethylammonium triflate [2-Sema][TfO]	Highly Brønsted-acidic ionic liquid assisted proton transport mechanism NMR spectra helps to investigate proton exchange through interaction between polar groups and water, proving the formation of hydrogen bonds in the polymer chain	10 mS/cm at 100 °C	[146]
1-butyl-3-metylimidazolium bis(trifluoromethane sulfonyl)imide [BMIm][TFSI]	The ionic conductivity of polymer increased with increasing ionic liquid percentage Prepared composites showed thermal stability in the range of 250–300 °C, with only 10% of weight loss when the temperature was higher than 350 °C LSV experiment indicated potential 4.85–5.45 V, suitable for high-energy battery application	8.8 × 10^−3^ S/cm at 55 °C	[147]
1-butyl-3-methylimidazolium dihydrogen phosphate (BMI-DHPH)	BMI-DHPH enhanced phosphoric acid absorption, thus increasing proton conductivity Cage-liked cross-linked polymer strengthened the mechanical Properties, which meet the approval level of tensile strength for HT-PEMFC application and improve ionic liquid retention	0.133 S/cm at 160 °C	[118]
1-metylimidazole trimethoxysilan	Ionic-liquid-functional silica enhanced the membrane performance Prepared membrane had better mechanical properties, higher proton conductivity due to the high ability to absorb phosphoric acid	0.106 S/cm at 170 °C	[148]
1-(3-trimethoxysilylpropyl)-3-methylimidazolium chloride	Hydrolysis of the ionic liquid forms a Si-O-Si network, improving the level of phosphoric acid doping and proton conductivity Si-O-Si network also improved the mechanical strength, chemical, and thermal stability	0.061 S/cm at 180 °C	[82]
Poly[1-(3*H*-imidazolium)ethylene] bis(trifluoromethanesulfonyl)imide	Polymeric ionic liquids play a significant role in enhancing mechanical strength and proton transfer	50 mS/cm at 200 °C	[149]
1,6-di(3-methylimidazolium)hexane bis (hexafluorophosphate) 1-butyl-3-methylimidazolium hexafluorophosphate	Introduction of dicationic ionic liquid enhanced the performance of the fuel cell Dicationic ionic liquid also increased membrane conductivity as it provides a workable ionic network for proton transfer	81 mS/cm at 180 °C	[150]

## Data Availability

Not applicable.

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
