# Peer review of "Ionic Liquid in Phosphoric Acid-Doped Polybenzimidazole (PA-PBI) as Electrolyte Membranes for PEM Fuel Cells: A Review"

_membranes, 2021, doi:10.3390/membranes11100728_

Round 1
Reviewer 1 Report
A very carefully prepared review, congratulations!
There are a few remarks, mostly editorial.
FIGURES FROM REFERENCES: Are copyright permissions included?
Page 1. Abstract: condensation to phosphate groups REMARK: What do you mean by condensation to phosphate groups?
the use of ionic liquid material on PA-PBI membranes CORRECT the use of ionic liquid material in PA-PBI membranes
Page 1: Sox that are harmful CORRECT SOx that are harmful
Page 2: higher tolerance of CO - CORRECT: higher tolerance to CO
Page 3: The most efficient method to overcome this problem is introducing ionic liquids into the polymer [9]. BETTER: The most efficient method to overcome this problem is introducing ionic liquids into the polymer phase [9].
Page 3: applied Nafion® membrane (Figure 2) REMARK applied Nafion® (Figure 2) membrane – The Figure does not show the membrane
Page 4: The PBI monomer is a macromolecule that contains aromatic linear heterocyclic REMARK: PLEASE READ IT CAREFULLY AND CORRECT
Page 12: since they only contain ions BETTER since they contain ions only
Page 17: Javanbakht et al. [104] used 1,3-di(3-methylimidazolium) propare dibromide dicationic ionic liquid CORRECT: Javanbakht et al. [104] used 1,3-di(3-methylimidazolium) to propare dibromide dicationic ionic liquid
Page 17 pr(mim)2Br2 BETTER propane dibromide dicationic ionic liquid (pr(mim)2Br2)
Page 17: Liu et al. [107] were prepared a series CORRECT: Liu et al. [107] prepared a series
Page 18: H2PO4- REMARK: wrong position of the minus sign. It should be written in a higher index
Author Response
"Please see the attachment."

Reviewer 2 Report
This manuscript proposes ionic liquid in phosphoric acid-doped polybenzimidazole (PA-PBI) as electrolyte membranes for PEM fuel cells. The topic is interesting, and certainly consistent with the contents to be proposed to the readers of “Membranes”. Moreover, the manuscript is well written and can be read with pleasure: this represents an important aspect in the current scenario of publications in international journals. Overall, I think that this manuscript has to be accepted, but the Authors should take into account the following minor revisions (in terms of bibliographic updates, grammar corrections and content deepening):
- It is difficult to review a manuscript without page numbers!!!
- Detailed revisions: I spent several hours reading this manuscript, and Authors are asked to follow carefully the attached PDF file where I highlighted some points to be addressed. The attached file also contains language mistakes and typos; some questions related to manuscript contents could also be present and Authors must consider them properly before submitting the revised manuscript. A point-by-point reply is required when the revised files are submitted.
- The Introduction should give a wider overview on the present scenario related to current trends in membranes-based technologies, both in terms of recently published reviews and research articles. In particular, membranes used in other energy conversion and storage devices are missing and a paragraph on this topic is highly suggested to be added in the Introduction. Authors are invited to go through the literature published in the last six months on these issues, and also on concepts developed some years ago in this field. Some of them are also mentioned in the above mentioned PDF file.

Author Response
"Please see the attachment."

Reviewer 3 Report
Summary:
The authors report a summary in using the ionic liquid material on phosphoric acid-doped polybenzimidazole (PA-PBI) membranes for polymer electrolyte membrane fuel cells. The ionic liquid preparation and the effect on the electrochemical and material performance are discussed.
1. The labels of plots in figures are randomly given in different formats.
2. For some reasons, several figures are cut and cannot be read.
3. In the section reporting Ionic Liquids in PBI Membranes (section 4), many paragraphs are focused on one certain reference. This makes this review as a case study and not an organized summary.
4. The electrochemical analysis condition and the testing parameters that would affect the device performance should be presented in the review and should be commented.
5. The reference section is not organized. Many references have incomplete publication information.
6. Remove the sentence directly copy from “A polybenzimidazole/graphite oxide based three layer membrane for intermediate temperature polymer electrolyte membrane fuel cells”
7. Remove the sentence directly copy from “Synthesis, characterization and application of a non-flammable dicationic ionic liquid in lithium-ion battery as electrolyte additive”
8. Table 4 is mainly copy from “A review of proton exchange membranes based on protic ionic liquid/polymer blends for polymer electrolyte membrane fuel cells” with no changes. This is not good.
9. The manuscript should be rewritten to remove all writing that are copied from other papers.
10. What is the difference between this review and a reported review: 10.1016/j.jpowsour.2020.229197?
Author Response
"Please see the attachment."

Reviewer 4 Report
The authors describe the improvement of the performance of PA-PBI membranes by the addition of ionic liquid compounds. The presented data are important in the development of fuel cell technologies. The manuscript merits publication. However, the authors asked to respond to the following comments:
- The authors need to clarify the presentation of figures:
- unclear markings in drawings: somewhere with parentheses, with one parenthesis, or no parentheses at all, i.e. "a", "(a", "(a)" (Figures 3, 5, 9,11, 12, 13). The authors need to use one style;
- Figure 6 lacks the "(c)" notation
- Figure 9 - double and unclear markings
- Figure 10 is not marked (b). The drawing does not fit on the page.
- Figure 11 - proofreading class - “(c))“ !!. I can't find a single drawing notation in general (g?).
- Figure 2 - the Nafion scheme should be cited.
- The equations in the drawings do not meet with those quoted in the sources cited.
- Figure 3: according to ref. [3] there are no equations like in part (b) at all. The equation looks different (no H2O and + additional equations) as in the ref. [35].
- Check Figure 4 [34] and Figure [45] for accurate presentation.
Author Response
"Please see the attachment."

Round 2
Reviewer 3 Report
The authors have addressed my comments.